# Complement Activation and Up-Regulated Expression of Anaphylatoxin C3a/C3aR in Glioblastoma: Deciphering the Links with TGF-β and VEGF

**DOI:** 10.3390/cancers15092647

**Published:** 2023-05-07

**Authors:** Franck Ah-Pine, Axelle Malaterre-Septembre, Yosra Bedoui, Mohamed Khettab, James W. Neal, Sébastien Freppel, Philippe Gasque

**Affiliations:** 1Unité de Recherche EPI (Études Pharmaco-Immunologiques), Université de La Réunion, Allée des Topazes, 97405 Saint-Denis, France; 2Service d’Anatomie et Cytologie Pathologiques, CHU de La Réunion, Avenue François Mitterrand BP450, 97448 Saint-Pierre, France; 3Service d’Oncologie Médicale, CHU de La Réunion, Avenue François Mitterrand BP450, 97448 Saint-Pierre, France; 4Institute of Life Sciences, Swansea Medical School, Sketty, Swansea SA2 8PY, UK; 5Service de Neurochirurgie, CHU de La Réunion, Avenue François Mitterrand BP450, 97448 Saint-Pierre, France; 6Laboratoire d’Immunologie Clinique et Expérimentale ZOI (LICE OI), CHU de La Réunion, Allée des Topazes, 97405 Saint-Denis, France

**Keywords:** complement, anaphylatoxins, C3aR, brain tumors, macrophages, TAM, glioblastoma, angiogenesis

## Abstract

**Simple Summary:**

The role of complement in cancer has raised much attention over the last decade. Recent studies have pointed to the tumor-promoting effects of complement activation within the tumor microenvironment of various cancers. The aim of our study was to assess the expression of the complement anaphylatoxin C3a and its receptor C3aR in the context of primary and metastatic brain tumors. We observed a high expression of C3aR in Grade 4 diffuse gliomas, i.e., glioblastoma multiforme, *IDH*-wildtype (GBM) and astrocytoma, *IDH*-mutant, Grade 4, compared to less aggressive primary brain tumors and brain metastasis. In GBM, C3aR was detected in tumor-associated macrophages, which also expressed VEGF, suggesting pro-angiogenic and tumor growth-supporting activities. In addition, robust levels of C3a were detected in GBM. Lastly, our results pointed to the critical role of TGF-β in the regulation of C3a and C3aR expression. Besides providing novel insights into glioma pathogenesis, our findings open new therapeutic avenues (C3aR antagonists) for this lethal disease.

**Abstract:**

The complement (C) innate immune system has been shown to be activated in the tumor microenvironment of various cancers. The C may support tumor growth by modulating the immune response and promoting angiogenesis through the actions of C anaphylatoxins (e.g., C5a, C3a). The C has important double-edged sword functions in the brain, but little is known about its role in brain tumors. Hence, we analyzed the distribution and the regulated expression of C3a and its receptor C3aR in various primary and secondary brain tumors. We found that C3aR was dramatically upregulated in Grade 4 diffuse gliomas, i.e., glioblastoma multiforme, *IDH*-wildtype (GBM) and astrocytoma, *IDH*-mutant, Grade 4, and was much less expressed in other brain tumors. C3aR was observed in tumor-associated macrophages (TAM) expressing CD68, CD18, CD163, and the proangiogenic VEGF. Robust levels of C3a were detected in the parenchyma of GBM as a possible result of Bb-dependent C activation of the alternative C pathway. Interestingly, in vitro models identified TGF-β1 as one of the most potent growth factors that upregulate VEGF, C3, and C3aR in TAM (PMA-differentiated THP1) cell lines. Further studies should help to delineate the functions of C3a/C3aR on TAMs that promote chemotaxis/angiogenesis in gliomas and to explore the therapeutic applications of C3aR antagonists for brain tumors.

## 1. Introduction

The complement (C) system comprises several innate immune molecules that sense and fight invasive pathogens (e.g., viruses and bacteria). Several recent studies have provided new perspectives on its function in the extravascular and interstitial tissue compartment [1,2,3,4]. The C is activated by three major pathways, all leading to the formation of C3 convertase complexes which cleave C3 molecules to C3a, one of the two major immunoregulatory C anaphylatoxins, and to C3b, a potent opsonin. The binding of C3b molecules to the surface of apoptotic/necrotic/microbe/cancer cells in an opsonization process makes them more appetizing for phagocytosis by macrophages [2,5]. Natural killer (NK) cells involved in cancer cell cytotoxicity, as with macrophages, use the C3b/inactivated C3b receptor called C receptor type 3 (formed by two subunits, α chain CD11b/Mac1, and α2 chain CD18) to recognize C-opsonized pathogens [6]. As the final step in the cytolytic pathway, components of the C terminal pathway form the membrane attack complex (MAC, i.e., C5b9) to provoke pore-forming lysis of the target cell [7]. To prevent overreactive C attack, several C inhibitors are either secreted (e.g., C1 inhibitor, factor H) or expressed at the cell membrane (e.g., CD46, CD55) [8]. Critically, cancer cells can regulate C attack by expressing high levels of several C inhibitors (e.g., CD46, CD55, CD59) [8,9]. 

C3a and C5a anaphylatoxins signal through transmembrane G protein-coupled receptors, C3aR and C5aR (CD88), respectively, to activate the recruitment of macrophages, neutrophils, mast cells, basophils, and eosinophils [10,11,12,13]. Anaphylatoxins binding to C3aR and C5aR can induce innate immune cell degranulation that releases cytokines and neuromediators (histamine, prostaglandins), which cause vasodilation, increase vascular permeability, and enhance neutrophil extravasation and further leukocyte chemotaxis. These events are likely to contribute to innate and adaptive immune responses against an infectious challenge or in response to tissue injuries (including cancer) to promote clearance of cell debris such as necrotic cells. 

We and others have shown that the C system is crucial in brain tissue homeostasis and pathologies [14,15,16,17,18,19]. Local C biosynthesis and expression of C receptors by glial cells (astrocytes and microglia) can clear host cell debris (including by synapse pruning) and promote the recruitment of stem cells to support repair. Interestingly, it has been shown that neural crest stem cells are attracted via C3a/C3aR [20], providing a novel role for C3a in vertebrate development and tissue repair [21]. C3a is also involved in cell proliferation and differentiation during liver regeneration, bone development, hematopoietic stem cell migration, and engraftment during hematopoiesis [11,22]. The same C components, receptors, and regulators are also abundantly expressed in the brain, particularly during an infectious/inflammatory response [2]. 

The ambivalent role of C in brain tumor growth is less well known and remains to be elucidated [23]. In non-brain tumors, C has been shown to be activated in the tumor microenvironment, representing an essential component of the chronic inflammatory response involved in various stages of tumorigenesis and the control of cancer progression [24,25]. 

Glioblastoma, *IDH*-wildtype (GBM) is the most malignant form of glioma, assigned to Grade 4 by the World Health Organization (WHO) classification of central nervous system (CNS) tumors [26]. Many studies have indicated the existence of abundant tumor-associated macrophages (TAM), derived either from blood-derived monocytes recruited to the CNS or from activated microglia within brain tissue/tumors. However, the exact mechanisms that drive macrophage/microglia infiltration into human CNS tumors and their potential contribution to tumor growth have yet to be identified [27]. While TAMs are capable of causing tumor cytotoxicity, they can also stimulate tumor growth through the secretion of growth-inducing factors (e.g., Transforming Growth Factor-β (TGF-β, Vascular Endothelial Growth Factor (VEGF), Interleukin 10 (IL-10), Tumor Necrosis Factor α) to promote angiogenesis, tumor immunosuppression, and metastasis, as demonstrated in several animal models [28,29]. These findings suggest an important role for TAM in GBM microvascular proliferation and high invasiveness, two features that define GBM. The critical role of C3aR + TAM in tumor progression, angiogenesis, and metastasis formation has been demonstrated in B16 melanoma and 3-methylcholanthrene (MCA) fibrosarcoma tumor models [30,31,32]. The evidence showing C3a binding to C3aR as a major chemoattractant for the recruitment of TAM into GBM adds to the importance of TAM for influencing human brain tumor behavior [33]. 

In this study, we found that GBM, *IDH*wt contained high numbers of TAM (CD68+, CD163+) that also expressed C3aR and VEGF. C3aR was also highly expressed in astrocytomas, *IDH*-mutant, Grade 4+. Fewer C3aR TAM were observed in lower-grade diffuse gliomas (WHO Grade 2–3). GBM were strongly immunostained for C activation pathway proteins, including C3c and alternative complement factor Bb. In vitro exposure to TGF-β increased C3, C3aR, and VEGF expressions in human TAM, but not in the U251MG glioma cell line. We suggest that C3aR macrophages/microglia, exposed to C3a, growth factors (TGF-β), and cytokines expressed in the brain tumor microenvironment, polarize towards a TAM phenotype and express VEGF, thus promoting angiogenesis and tumor growth.

## 2. Materials and Methods

### 2.1. Tissue Samples

Tissue samples were retrieved from the tumor tissue bank of the teaching hospital (CHU) of La Réunion. We tested 37 formalin-fixed, paraffin-embedded (FFPE) brain tumors, including GBM, ***IDH***wt, Grade 4 (GBM, *n* = 5), astrocytomas, *IDH*-mutant, Grade 2 (A2, *n* = 5), Grade 3 (A3, *n* = 5), and Grade 4 (A4, *n* = 5), oligodendrogliomas, *IDH*-mutant and 1p/19q-codeletd, Grade 2 (O2, *n* = 5) and Grade 3 (O3, *n* = 5), classic ependymomas, Grade 2 (*n* = 5), and metastatic carcinoma brain tumors (*n* = 2), diagnosed between 2018 and 2020. Implied consent was obtained, in compliance with French regulations, and as part of the tumor tissue bank of La Reunion (French government registry DC-2016-2860). Integrated diagnoses, according to the 2021 WHO classification of CNS tumors, were confirmed by a trained pathologist (FA). 

### 2.2. Immunohistochemistry Experiments

FFPE sections were cut at 4 μm and processed on the Leica Biosystems Bond-III automated staining system for immunohistochemical (IHC) investigations, according to the manufacturer’s instructions (Leica Biosystems, Wetzlar, Germany). In all 37 brain tumors, we evaluated the expression of C3aR (anti-C3aR, clone D12, dilution 1:200, Santa Cruz, Dallas, TX, USA). In 5 GBM, we investigated the expression of CD45 (anti-CD45, clone X16/99, ready-to-use, Leica), CD68 (anti-CD68, clone 514H12, ready-to-use, Leica), CD31 (anti-CD31, clone 1A10, ready-to-use, Leica), CD34 (anti-CD34, clone QBEnd/10, ready-to-use, Leica), CD1a (anti-CD1a, clone MTB1, ready-to-use, Leica), Ki67 (anti-Ki67, clone MM1, ready-to-use, Novocastra, Leica, Wetzlar, Germany). Revelation was carried out using the BOND polymer refine detection kit (Ref Ds9800, Leica).

For double immunostaining, we performed double sequential IHC staining on the Leica Biosystems Bond-III automated staining system, to evaluate the expression of C3aR (anti-C3aR, clone D12, dilution 1:200, Santa Cruz), CD163 (anti-CD163, clone 10D6, dilution 1:100, Leica), CD68 (anti-CD68, clone 514H12, ready-to-use, Leica), C3a (anti-C3a, clone K13/16, dilution 1:200, Biolegend, San Diego, CA, USA), CD248 (anti-CD248, clone E9Z7O, dilution 1:200, Cell Signaling, Danvers, MA, USA), and ERG (anti-ERG, clone EPR3864, ready-to-use, Ventana, Singapore). Revelation was carried out using the BOND polymer refine detection kits (Ref Ds9390 and Ds9800).

To detect complement activation, we used anti-C3 (L440, in-house, dilution 1:100), anti-C3c (polyclonal, dilution 1:100, ref. A0062, Dako, Glostrup, Denmark), anti-C3a (clone K13/16, dilution 1:200, Biolegend) and anti-fragment B (Bb) (kind gift of Pr Otto Götze, Germany, dilution 1:100). TGF-β was immunodetected using anti-TGF-β (clone FO311, dilution 1:100, Santa Cruz). 

### 2.3. Immunofluorescence Experiments

Frozen GBM samples were sectioned using a Leica™ CM 1950 cryostat (5 µm thick) at a chamber temperature of −15 °C. Frozen Sections were mounted on SuperFrost Plus microscope slides and fixed with frozen 95% ethanol for 5 min. For antibody labelling, sections were incubated with primary antibodies: CD18 (mouse antibody ref TS1/18 Biolegend, 1:100) and C3aR (rabbit antibody PA5109467, Ref 17107403, Fischer Sci., Hampton, NH, USA, 1:100) or C3aR (mouse antibody clone D12, ref SC133172, Santa Cruz, 1:100) and VEGF (rabbit antibody Millipore, Burlington, MA, USA, AB1442, 1:100), or C3aR (mouse antibody clone D12, ref SC133172, Santa Cruz, 1:100) and CD248/endosialin (rabbit antibody clone E9Z7O, Cell Signalling, 1:200), followed by Alexa Fluor 488-conjugated donkey anti-mouse (Invitrogen, Waltham, MA, USA, A21202, 1:1000 dilution) and Alexa Fluor 594-conjugated donkey anti-rabbit (Invitrogen A21207, 1:1000 dilution), counterstained with DAPI (Sigma-Aldrich, Roedermark, Germany) and mounted with Vectashield (Vector Labs, Burlingame, CA, USA; Clinisciences, Nanterre, France). Fluorescence was observed using a Nikon Eclipse E2000-U microscope (Nikon, Tokyo, Japan) and images were captured and processed using a Hamamatsu ORCA-ER camera and the imaging software NIS-Element AR (Nikon, Tokyo, Japan).

### 2.4. Cells and Stimulatory Conditions

In vitro human cell lines were used to address the regulated expression of C3a/C3aR by glioma cells and macrophages. The human glioma cell line U251MG (known to express the CD133 Glioma stem cell marker) [34] and the canonical macrophage cell line THP1 were obtained from ATCC and grown in Modified Eagle’s Medium or RPMI (PAN Biotech, Bayern, Germany), respectively. Both media were supplemented with 10% heat-inactivated fetal bovine serum (FBS; PAN Biotech, 3302 P290907), 2 mM of L-glutamine (Biochrom AG, K0282), 0.1 mg/mL penicillin-streptomycin (PAN Biotech, P0607100), 1 mM of sodium pyruvate (PAN Biotech, P0443100) and 0.5 µg/mL of amphotericin B (PAN Biotech, P0601001). 

THP1 were differentiated to macrophages with the phorbol ester PMA (10 ng/mL) for 72 h before treatments. We have previously shown that PMA-THP1 cells express high levels of C3aR [35]. IFN-γ, TGF-β1, Sonic Hedgehog (Shh), Wnt, and Platelet-derived growth factor-BB (PDGF-BB) were purchased from Peprotech (Cranbury, NJ, USA) and used at 20 ng/mL for 24h stimulation of U251MG and PMA-differentiated THP1.

### 2.5. Quantitative RT-PCR Analysis

Total RNA were extracted from harvested cells (cultured in 6-well plates) using a Zymo kit (ZYMO, Seattle, WA, USA, Catalog R1035). Then, 200 µL of RNA Shield and 800 μL of lysis buffer were added to each well, collected, and kept at −20 °C until use. qRT-PCR experiments were carried out using the One Step Bioline Sensifast Probe NO-ROX One step Kit (Meridian Bioscience, Cincinnati, OH, USA, Bio-76005) containing the Syber Green reagent (Lonza, Basel, Switzerland, Cat. No.50513). qRT-PCR was performed in a final volume of 5 μL containing 1 μL of extracted total RNA per reaction, 2.7 μL of enzyme mix, and 1.3 μL of primers mix with a final primer concentration of 250 nM. qRT-PCR was carried out in a Quantstudio 5 PCR thermocycler (Thermo Fisher Scientific, Waltham, MA, USA). Relative gene expression was calculated using GAPDH as a reference gene. Experiments were carried out in triplicate. Primer sequences related to the genes are listed in Table 1 below.

### 2.6. Statistical Analysis

Data were expressed as means ± SEM. All assays were performed in triplicate independent experiments. Statistical analysis was performed using Graph Prism 6 software. Significant differences (*p* < 0.05) between the means were determined by analysis of variance (ANOVA) procedures followed by a multiple comparison test (Bonferroni).

## 3. Results

### 3.1. C3aR Is Highly Expressed in Glioblastoma, IDHwt and in Astrocytoma, IDH-Mutant, Grade 4

In non-tumoral GBM-adjacent brain tissue, C3aR cells were scarce (Figure 1a,b). All WHO Grade 4 diffuse gliomas (i.e., five GBM and five A4) demonstrated abundant C3aR staining within the tumor, both within the viable (Figure 1c,d and Figure 2c) and the peri-necrotic areas. The C3aR cells had characteristic features of microglia/macrophages with an oval nucleus and plump cytoplasm terminating in a few processes, or showed an amoeboid-like appearance. The small size and regularity of the nucleus distinguished C3aR cells from the atypical and proliferating Ki67 malignant cells (Figure 3h). C3aR was not expressed in GBM/A4 cells (Figure 1d and Figure 2c).

### 3.2. C3aR Is Less Expressed in Other Brain Tumors

Among adult-type diffuse gliomas, A2, A3, and O2 also showed C3aR cells, although to a lesser extent than Grade 4 gliomas (Figure 2). The morphology of these cells was very similar to the resting ramified microglia, with a perinuclear cytoplasm and fine dendritic processes. In O3, classic ependymomas (Grade 2), and metastatic carcinoma tumors, we found scarce C3aR staining (Figure 2).

### 3.3. Tumor-Associated Macrophages Express High Levels of C3aR in GBM

Immunostaining of semi-serial FFPE sections of GBM revealed that the C3aR cells mostly localized at the perivascular level and also expressed macrophage markers, such as CD68 and CD45 (Figure 3a–c). We have previously shown that C3aR is expressed by endothelial cells in vitro and in brain meningitis cases [35,36]. The CD31 and CD34 staining, detected in the endothelial cells, was consistent with microvascular proliferation (angiogenesis), a typical histological feature of GBM (Figure 3d,e). Although there was histological evidence of angiogenesis, C3aR was not expressed by these reactive and proliferating endothelial cells. Pericytes, also known as perivascular mesenchymal stem cells (pMSC), expressing the smooth muscle cell actin (SMA) marker, were associated with angiogenic vessels, but these did not express C3aR (Figure 3f). CD1a staining, which would detect dendritic cells, was absent in the GBM specimen (Figure 3g). Ki67 staining revealed a high tumor cell proliferation index (Figure 3h), but Ki67+ cells were not stained for C3aR.

### 3.4. C3aR Tumor-Associated Macrophages Also Express VEGF in GBM

TAM are known to be polarized in either pro (M1) or anti-inflammatory/pro-angiogenic (M2) phenotypes, the latter being important in tumor development [37]. We next sought to verify C3aR expression on TAM in GBM cases by double staining using antibodies against several canonical macrophage markers. We found that C3aR colocalized to cells expressing α2 integrin (CD18) and CD68 (Figure 4a,c). CD163 is generally recognized as an M2 marker, and we identified C3aR TAM that also expressed CD163 in GBM cases. CD163-/C3aR TAM were also identified (Figure 4b).

Interestingly, C3aR TAM also strongly expressed VEGF (Figure 4d) and may correspond to the subset of F4/80high C3aRhigh TAM driving robust angiogenesis and metastasis observed in several mouse models of cancer [30,31,32]. Notably, we found that C3aR was also expressed by non-TAM cells (Figure 4a, white arrowhead) and may correspond to mast cells, known to express abundant levels of C3aR [38].

### 3.5. C3aR Is Not Expressed in Perivascular Mesenchymal Cells/Pericytes or Endothelial Cells

C3aR has been involved in the chemotaxis of mesenchymal stem cells (MSC) in vitro [39]. Brain pericytes originate from the neural crest-derived MSC and have been shown to express C3aR to control neurogenesis [20]. In GBM, we used an anti-CD248 (endosialin) to identify perivascular MSC/pericytes. Double staining (immunofluorescence and immunoperoxidase) experiments failed to indicate that CD248+ cells and CD90+ cells express C3aR (Figure 5a,b). ERG+ endothelial cells did not stain for C3aR (Figure 5c).

### 3.6. The Alternative Complement Pathway Is Activated to Generate C3a

C3a anaphylatoxin may be released in the tissue parenchyma through the actions of two main proteases: the C3 convertases (e.g., C3bBb of the alternative pathway and C4bC2b of the Classical and Lectin pathway) and cathepsin B and L [40]. Cathepsins are known to be abundantly expressed in brain cancer, contributing to GBM invasiveness [41]. Focusing on C activation, we found abundant C3a staining in GBM and robust staining for Bb neoepitope, a component of the alternative C pathway (Figure 6). Both examples of staining were localized to the brain parenchyma, while C3 and C3c staining were detected in the blood vessel lumen and areas of tumor cell necrosis, respectively (Figure 6a,b). Double immunostaining revealed that C3aR TAM were near C3a deposits (Figure 6e). We further found that GBM tumors were strongly stained for the immunoregulatory factor TGF-β (Figure 6f).

### 3.7. TGF-1β Is a Major Regulator of C3 and C3aR Expression in PMA-Differentiated THP1 Cell Line In Vitro

Our data support the paradigm that the GBM tumor environment may favor elevated expression of C components such as C3 and C3aR. However, the mechanisms of C3 and C3aR regulation in the context of GBM are still to be elucidated. Hence, we performed RT-PCR analyses to test for the effects of several growth factors and cytokines on C activation. Indeed, T cells infiltrating the tumors may contribute to this regulation through the release of IFN-γ, while other cell types of the parenchyma may express TGF-β1 (tumor cells), Shh (reactive astrocytes), Wnt (neurons), and PDGF-BB (proliferating endothelial cells) [42]. 

We found that the THP1 cell line, differentiated with PMA phorbol ester to model the TAM phenotype [35], expressed the highest transcript levels of C3, C3aR, Cathepsin L, VEGF, and CCL5/RANTES when compared to the U251MG GBM model (Figure 7). C3aR mRNA was 10–20 fold more expressed by PMA-THP1 when compared to U251MG. TGF- β1 treatment for 24h was the optimal condition to upregulate dramatically and significantly the expression of all transcripts (Figure 7). Moreover, IFN-γ, to a lesser extent, was also able to significantly modulate the expression of all transcripts by PMA-differentiated THP1 except for CCL5/RANTES. Of note, U251MG expressed high levels of CCL2/MCP1 mRNA, particularly in response to TGF-β1 (Figure 7).

## 4. Discussion

The complement system is expressed in a variety of tumor types, including glioblastomas, and recent studies have uncovered the expression patterns of C components in gliomas [23,43]. Indeed, C activation in gliomas was evidenced by the detection of C4d by immunohistochemistry in diffuse gliomas, while high C4d staining intensity was correlated with poor prognosis [44]. In addition, Bouwens et al. have described abundant C1q and C3 expression in both necrotic and non-necrotic areas of GBM tissues by immunohistochemistry [45]. As a result of C activation, high expression of *C1QA*, *C1QB*, and *C1QC*, encoding the three chains of C1q, were observed in gliomas [46]. Additionally, Patel et al. found that glioblastoma cells were able to express several genes of the complement pathway, including C3 [47]. Interestingly, a recent study showed that MSCs in human glioblastoma are able to secrete C5a, enhancing the invasion of GBM cells [48]. While these results concur with the activation of C in gliomas, its exact role in gliomagenesis, as elegantly debated recently by Van der Vlis in 2018, remains to be fully elucidated [26].

Herein, our data suggest an essential role of C3a/C3aR expressed particularly by TAM in Grade 4 diffuse gliomas, i.e., GBM and astrocytomas, ***IDH***-mutant, Grade 4, to mediate monocyte/macrophage chemotaxis to the brain. Our immune phenotyping experiments confirmed that C3aR is expressed in macrophages (CD163+, CD18+, and CD68+) but not in MSC-like pericytes (identified as CD248+/CD90+, αSMA+ cells). C3aR expression has been reported on endothelial cells, but in human GBM, we did not find detectable levels of C3aR on ERG+ endothelial cells.

The activation of TAM (from blood monocytes or resident microglia) is known to promote polarization towards anti-inflammatory and proangiogenic phenotypes and contribute to tumor growth. We observed that C3aR density was notably elevated in high-grade tumors, suggesting a supportive role for TAMs in glioma progression. Several factors such as TGF-β1 and IFN-γ may positively control the level of C3aR expression and may be produced in less quantities in metastatic and low-grade gliomas. The expression of VEGF by human TAM (C3aR) connects this population of macrophages with the promotion of microvascular proliferation, a defining feature of GBM [49]. 

Focusing on monocyte/macrophage chemotaxis, TAMs constitute up to 30% of all immune cells within glioblastoma [37]. The recruitment of TAMs into glioma is known to be mediated by cytokines and chemokines released by tumor cells and cancer-associated fibroblasts (CAF). These factors have been extensively studied and include CC-chemokine CCL2/MCP1, CCL7, glial cell line-derived neurotrophic factor (GDNF), VEGF, ATP (released from necrotic cells), macrophage colony-stimulating factor 1 (CSF1), granulocyte–macrophage colony-stimulating factor (GM–CSF) and periostin [27,50]. Our study argues for a positive role of the chemoattractant C3a, abundantly produced (through the cleavage of C3) by tumor cells and TAM, as evidenced by our U251MG and THP1 cell model systems. C3 is the precursor of C3a and has been detected at the primary tumor site in the serum of several solid cancers and is associated with poor prognosis in colorectal and ovarian cancer [25,51]. We and others have previously shown in vitro that several glioblastoma cell lines (T98G, CB193, U105MG) produce abundant levels of C3 (as well as factor B and factor D), particularly in response to T/NK cell-derived IFN-γ [52,53]. Herein, we used the U251MG and confirmed that C3 expression was upregulated in response to IFN-γ. While comparing U251MG and PMA-differentiated THP1, we found that the macrophage cell line was expressing C3 transcript at a much higher level, and this expression was dramatically and significantly upregulated by TGF-β1, a factor known to be abundantly expressed by GBM tumor cells [54] as well as microglia [55]. 

High levels of C3c (cleavage product of native C3) were associated with necrotic cells and may correspond to the activation of the classical pathway mediated by the binding of C1q to extracellular DNA [56]. In GBM, we observed a strong expression of Bb, which may indicate activation of the complement system via the alternative pathway. This was particularly prominent in perivascular regions that also exhibited high expression of C3a. High levels of C3a in GBM may also result from C3 cleavage by other cell-derived proteases such as cathepsins [40]. Cathepsin L is known to play an important role in malignant gliomas [57]. Interestingly, we found high levels of cathepsin-L expression in our model of TAM in response to TGF-β1 and IFN-γ. Shh, Wnt, and PDGF-BB also strongly upregulated cathepsin-L expression. 

In addition to chemotaxis, C3a may contribute to the polarization of the newly recruited C3aR monocytes towards a non-inflammatory phenotype with important immunosuppressive immunoregulatory and tissue repair functions [58]. We found that C3aR was highly expressed by perivascular macrophages of both subtypes, CD163 negative and CD163 positive (canonical M2-like marker) in GBM. We suggest that the expression of C3aR by TAMs GBM may also contribute to immunosuppression and promote tumor growth, as observed in various studies using melanoma and sarcoma mouse models. For instance, Nabizadeh and colleagues reported that B16 melanoma-TAM numbers were reduced in C3aR deficient animals compared to control wild-type animals [33]. In a more recent study, Davidson and colleagues observed that the neutralization of C3a (using a blocking anti-C3a or a pharmacological antagonist of C3aR) leads to a reduction of the tumor mass and is associated with an increase in non-differentiated pro-inflammatory Ly6C+ subpopulations of monocytes [30]. These findings support the critical role of C3a in monocyte recruitment and differentiation to anti-inflammatory macrophages. C3aR expression was also confirmed in a study that used the MN/MCA1 fibrosarcoma or 3-MCA-derived tumor models. The authors demonstrated that C3aR was mainly expressed by tumor-infiltrating macrophages, monocytes, neutrophils, and to a much lower extent, by T cells [32]. In contrast, no C3aR expression was observed in tumor cells suggesting that a direct effect of C3a on tumor cells was unlikely. Interestingly, this study also confirmed a significant reduction of macrophages recruited into the tumor bed of C3 and C3aR KO compared to WT MN/MCA1 tumor-bearing mice. Again, the frequency of Ly6C+ inflammatory monocytes was higher in C3aR−/− mice than in WT mice. The authors reported an upregulation of M1-like markers (CD11c, major histocompatibility complex class II (MHC-II), CD80, and CD86) and a reduction in the frequency of CD206+ (an M2-like marker) TAMs in C3aR-deficient mice compared to WT. C3aR deficiency in mice had a major impact on T-cell-dependent antitumor effector mechanisms, and the frequency of tumor-infiltrating CD4+ T lymphocytes was significantly increased in C3aR-deficient mice to mediate a more robust adaptive immune response against cancer [32]. The frequency of CD8+ tumor-infiltrating T cells was also higher in C3aR KO [30]. The expression of C3aR by TAM has relevance for clinical progression and survival. Magrini et al. further validated that C1q, C4d, and C3c deposition and C3aR expression were elevated in a cohort of human undifferentiated pleomorphic sarcoma (UPS). High C3aR immunoreactivity was detected in 14 of 19 UPS samples and confined to infiltrating macrophages. More importantly, disease-free and metastasis-free survival were higher in C3aR neg than C3aR + UPS patients, indicating that C3aR expression contributed to tumor aggressiveness and mirroring findings observed in the mouse models mentioned above [32]. 

Within GBM, we propose that TAM expressing C3aR and contributing to a C3a-dependent autocrine signaling mechanism have important proangiogenic functions. We found that C3aR TAM expressed high levels of VEGF. Interestingly, these VEGF+ C3aR+ TAM may correspond to the subset of F4/80high C3aRhigh TAM driving robust angiogenesis and metastasis, as reported in several mouse models of cancer [30,31,32]. In comparison with F4/80low TAMs, F4/80 high TAMs expressed higher levels of C3aR (as well as C5aR/CD88) and the macrophage colony-stimulating factor (M-CSF) receptor (also known as CSF1R, CD115), CD206, CCR5 (receptor for RANTES/CCL5) and the angiopoietin receptor, Tie2, known to be typically expressed by a proangiogenic mono-macrophage subset [59]. F4/80 marker can discriminate the different populations of TAM, although its use is restricted to mice. Further immunohistochemical analyses are now highly warranted to better identify the sub-phenotypes of C3aR TAM in human GBM and their association with either M2 or M1 markers. 

Little is known about the capacity of C3a to modulate VEGF expression in tumor settings directly. While exogenous C3a has been shown to significantly upregulate VEGF mRNA levels in retinal epithelial ARPE-19 cell models, experiments are warranted to analyze this response using macrophage or microglia cell lines [60]. 

## 5. Conclusions

Persistent immune responses involving the C system may exacerbate the recruitment and polarization of innate immune cells in neoplastic microenvironments. High expression of TGF-β produced by TAM together with IFN-γ (derived from recruited T lymphocytes/NK cells) in GBM have potential effects on TAM chemotaxis by upregulating C3aR and C3a genesis. Furthermore, C3aR TAM in GBM is well-placed to regulate pro-angiogenic and pro-survival pathways that potentiate tumor growth and tissue infiltration. It should be possible to isolate the C3aR TAM from the GBM cases and to perform gene profiling studies. These experiments are warranted to begin to dissect the emerging role of the C3a/C3aR signaling pathway in the general mechanisms of the innate immune response to brain tumors. Current trials investigating the efficacy of C inhibitors, such as compstatin and its derivatives which prevent C3 cleavage, may be of interest for tumor therapy [61]. It should be possible to combine these treatments with pharmacological antagonists of C3aR. Owing to the critical role of C in immune regulation, these therapies could be combined with checkpoint inhibitors to co-modulate the adaptive immune response against tumor progression.

## Figures and Tables

**Figure 1 cancers-15-02647-f001:**
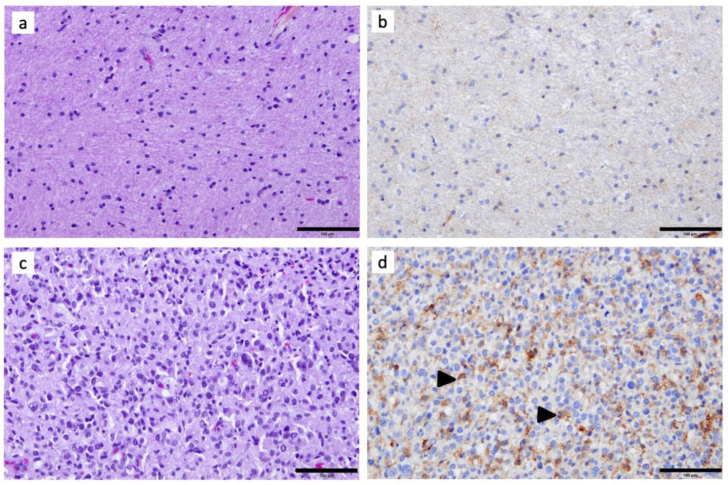
Immunohistochemical assessment of C3aR in a representative case of GBM: (**a**) Hematoxylin–Phloxine–Safran and (**b**) anti-C3aR staining (clone D12, dilution 1:200, Santa Cruz) of adjacent non-tumoral brain tissue. (**c**) Hematoxylin–Phloxine–Safran (×200) and (**d**) anti-C3aR staining of GBM tissue. Images are shown at ×200 magnification. The bar indicates 100 μm size. Arrowheads point to examples of C3aR cells.

**Figure 2 cancers-15-02647-f002:**
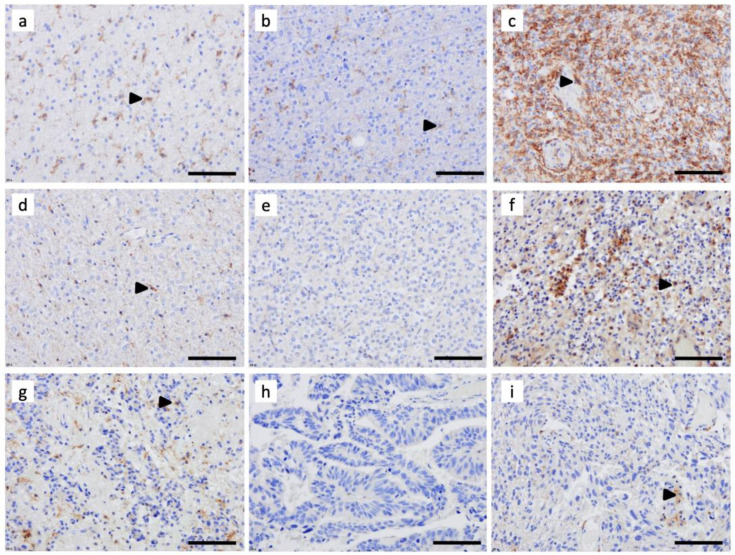
Immunohistochemical assessment of C3aR expression in (**a**) astrocytoma *IDH*-mutant, Grade 2, (**b**) astrocytoma *IDH*-mutant, Grade 3, (**c**) astrocytoma *IDH*-mutant, Grade 4, (**d**) oligodendroglioma *IDH*-mutant and 1p/19q-codeleted, Grade 2, (**e**) oligodendrogliomas *IDH*-mutant and 1p/19q-codeleted, Grade 3, (**f**) glioblastoma *IDH*wt, Grade 4, (**g**) classic ependymoma, Grade 2, and (**h**,**i**) two brain metastatic carcinomas. FFPE sections were cut at 4 μm and processed on the Leica Biosystems Bond-III automated staining system for immunohistochemical (IHC) investigation using anti-C3aR (clone D12, dilution 1:200, Santa Cruz). Revelation was carried out using the BOND polymer refine detection kit (Ref Ds9800, Leica). Images are shown at ×200 magnification. The bar indicates 100 μm size. Arrowheads point to examples of C3aR cells.

**Figure 3 cancers-15-02647-f003:**
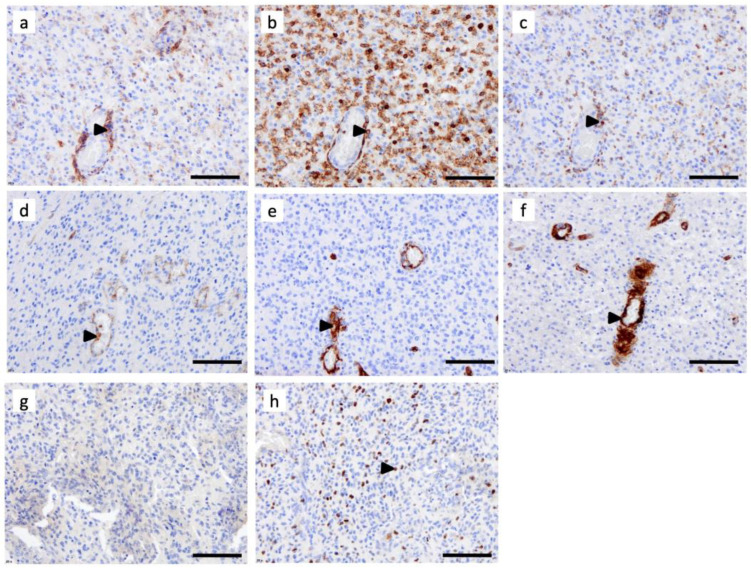
C3aR distribution is enriched in perivascular regions of glioblastoma multiforme, *IDH*-wildtype (GBM). Serial FFPE sections of a GBM *IDH*wt were stained using monoclonal antibodies for (**a**) C3aR, (**b**) CD45, (**c**) CD68, (**d**) CD31, (**e**) CD34, (**f**) SMA (**g**) CD1a, and (**h**) Ki67. Images are shown at ×200 magnification. The bar indicates 100 μm size. Arrowheads point to examples of positive cells.

**Figure 4 cancers-15-02647-f004:**
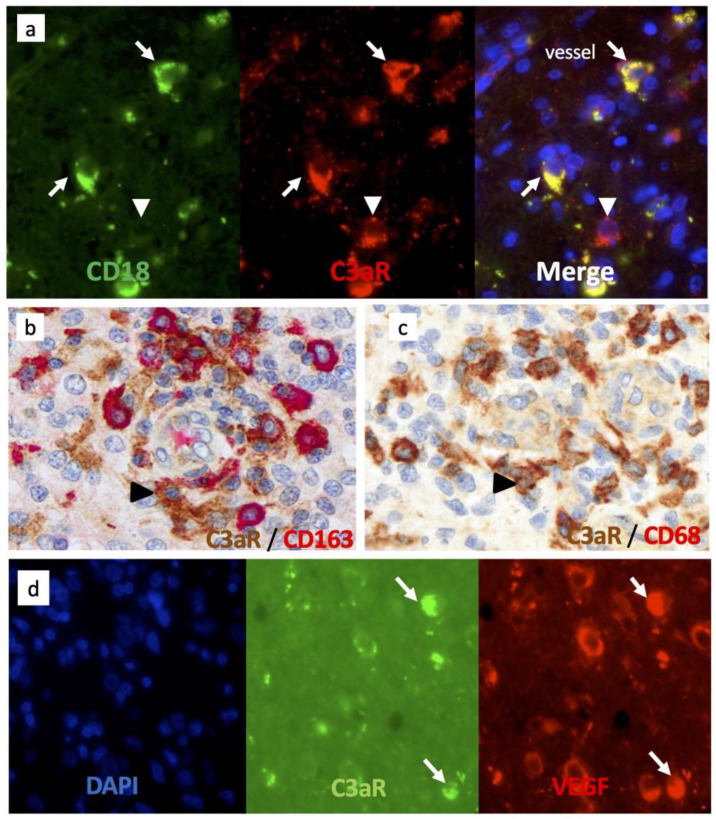
Tumor-associated macrophages/microglia express high levels of C3aR and VEGF in GBM. (**a**) Frozen tissue sections of GBM were sequentially stained for CD18 (mouse antibody (Mab), Green) and C3aR (rabbit Ab Thermo, Red), Alexa 488 or 594-conjugated secondary antibodies and counterstained with DAPI (to stain nuclei in blue). The arrows indicate C3aR/CD18+ cells while the arrowhead points to a mast-cell-like C3aR/CD18-cell. (**b**,**c**) Double staining of GBM paraffin wax sections for C3aR (Mab, clone D12 Santa Cruz) and either (**b**) anti-CD163 (Clone 10D6) or (**c**) Mab anti-CD68 (clone 514H12, Leica). Arrowheads point to C3aR cells. (**d**) Frozen tissue sections of GBM were double-stained for C3aR (Mab, clone D12), Green) and VEGF (rabbit Ab Millipore, Red), Alexa 488 or 594-conjugated secondary antibodies and counterstained with DAPI. Arrows point to TAM-like amoeboid cells double-stained for C3aR and VEGF. All images are shown at ×400 magnification.

**Figure 5 cancers-15-02647-f005:**
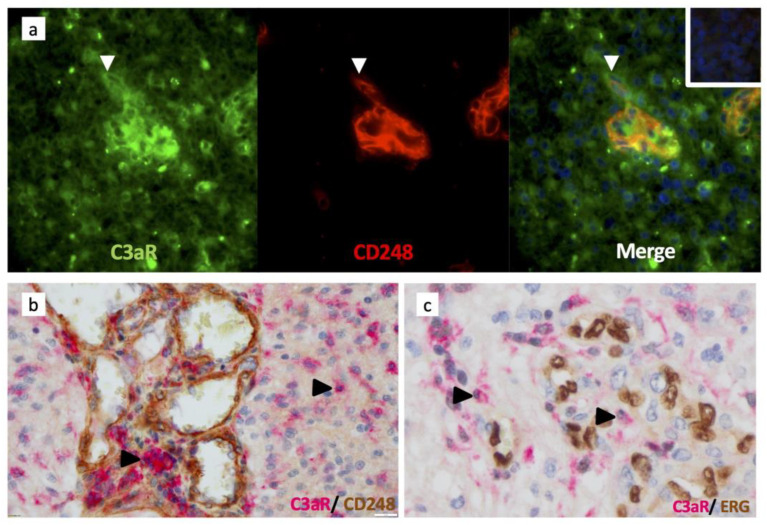
Pericytes/perivascular MSC and endothelial cells do not express C3aR in GBM. (**a**) Frozen tissue sections of GBM were sequentially double-stained for C3aR (mouse antibody (Mab D12), Green) and CD248/endosialin (rabbit Ab CST, Red), Alexa 488 or 594-conjugated secondary antibodies, and counterstained with DAPI (to stain nuclei in blue). Arrowhead indicates pMSC single-stained for CD248 and next to C3aR infiltrating TAM-like cell (×200). (**b**) Double immunoperoxidase staining of GBM paraffin wax sections for C3aR (Mab, clone D12 Santa Cruz) and rabbit anti-CD248 (×200). Arrowheads indicate examples of C3aR cells. (**c**) Double immunoperoxidase staining of GBM paraffin wax sections for C3aR (Mab, clone D12 Santa Cruz) and anti-ERG (clone EPR3864, Ventana) (×400). Arrowheads indicate examples of C3aR cells.

**Figure 6 cancers-15-02647-f006:**
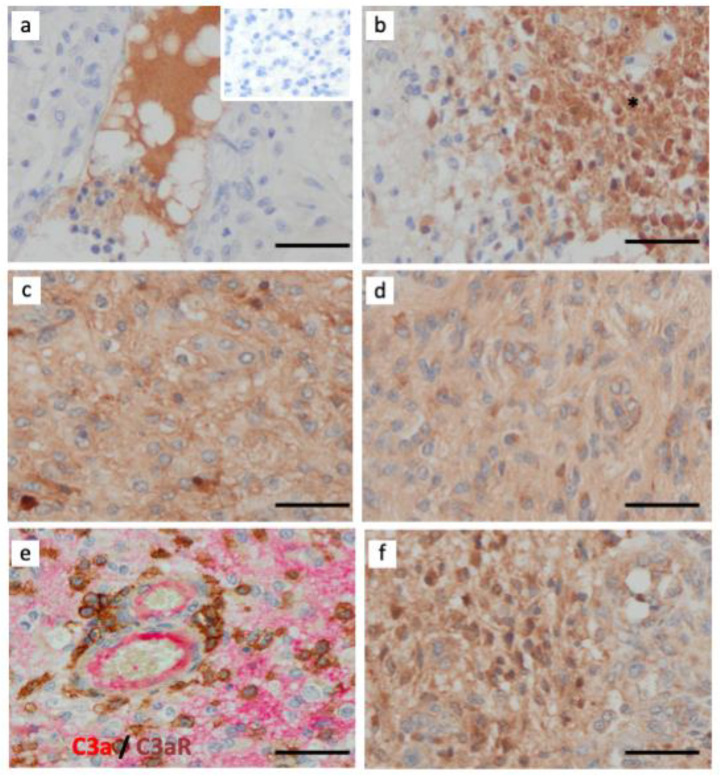
Activation of complement system via the alternative pathway (Bb-dependent) in GBM. Paraffin wax sections of GBM were single-stained for several components of the C system using (**a**) rabbit anti-native C3 (inset is the negative control of irrelevant antibody staining); (**b**) rabbit anti-C3c (fragment generated after C activation); (**c**) rabbit anti-anaphylatoxin C3a; (**d**) rabbit anti-Bb neoepitope (generated after activation of the alternative pathway); (**e**) Paraffin wax sections of a GBM case were double-stained for C3a (clone K13/16, red) and C3aR (clone D12, brown) and (**f**) rabbit anti-TGF-β. Immunoperoxidase and hematoxylin stains were applied. The bar indicates 50 μm. All images are shown at ×400 magnification.

**Figure 7 cancers-15-02647-f007:**
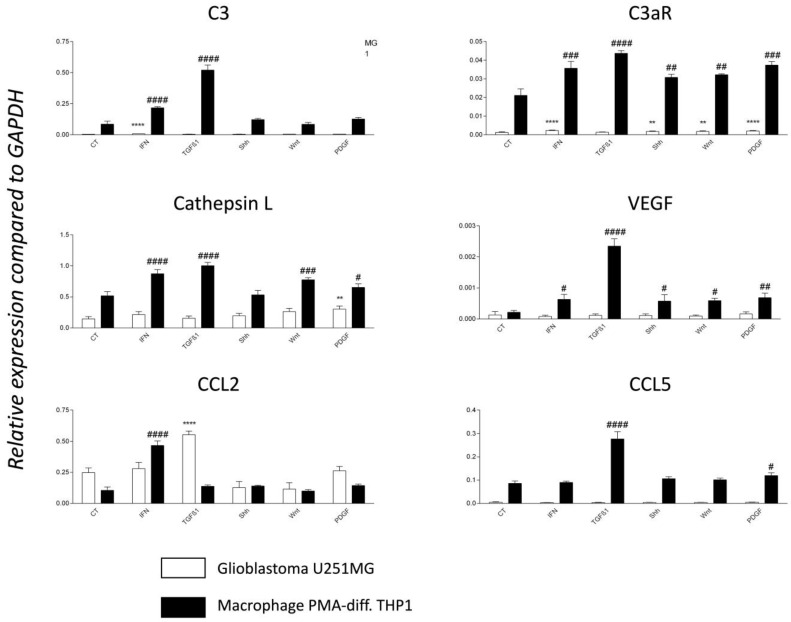
TGF-β 1 is a major regulator of C3 and C3a receptor mRNA expression. RT-PCR analyses of PMA-differentiated THP1 monocyte cell line (model of TAM) and glioblastoma cell line U251MG in untreated conditions (CT) or following treatment for 24 h with 20 ng/mL of either recombinant IFN-γ, TGF-β1, Shh, WNT or PDGF-BB. N = 4 independent experiments. Expression levels are expressed relative to the housekeeping gene GAPDH. Statistical analyses are indicated (* for U251MG, # for THP1) while comparing basal and treated conditions. Significance is indicated in the figure as follows: *p*-values ≤ 0.05 (#), *p*-values ≤ 0.01 (** or ##), *p*-values ≤ 0.001 (###) and *p*-values ≤ 0.0001 (**** or ####).

**Table 1 cancers-15-02647-t001:** Primers used for qRT-PCR analysis.

Gene	Forward Sequence (5′-3′)	Reverse Sequence (5′-3′)
*GAPDH*	TGCGTCGCCAGCCGAG	AGTTAAAAGCAGCCCTGGTGA
*CCL5/RANTES*	TCCTCATTGCTACTGCCCTC	TCGGGTGACAAAGACGACTG
*CCL2/MCP1*	CTGCTCATAGCAGCCACCTT	CTTGAAGATCACAGCTTCTTTGGG
*C3*	TCAACCACAAGCTGCTACCC	CTGGCCCATGTTGACGAGTT
*C3aR*	AGACAGGACTCGTGGAGACA	CTCAGCAGAGAAAGACGCCA
*Cathepsin L*	CTGGCAACGAGAGCGTCTAC	CCATCCTTCTTCATTCATGCCG
*VEGF-A*	ACAACAAATGTGAATGCAGACCA	GAGGCTCCAGGGCATTAGAC

## Data Availability

The data can be shared up on request.

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
