# Peer review of "Complement Activation and Up-Regulated Expression of Anaphylatoxin C3a/C3aR in Glioblastoma: Deciphering the Links with TGF-β and VEGF"

_cancers, 2023, doi:10.3390/cancers15092647_

Round 1
Reviewer 1 Report
This is a very interesting manuscript by Franck AH-Pine et.al, addressing complement C3a/C3aR in human glioblastoma patients. The authors presented, in glioblastoma patients, tumor associated macrophages stimulates higher levels of C3a and C3aR, which activate VEGF and cause tumor growth. Authors also proposed, complement therapy for glioblastoma. Overall, manuscript was well written with report data was good with figures. However, there are MANY PLACES in the manuscript in which the presentation must be improved as noted below.
Line 47: Graphical abstract title is missing
Line 48: Reference for “Complement (C) is not expressed nor activated in the healthy human brain”.
Line 48: Remove “sharp contrast”
Line 49: Should be release of “anaphylatoxin” C3a
Line 51: Remove “precursor (i.e., C3)” in line 51 and in appropriate sentences (line 349, 352, 369, 400) and it should be C3 or C3a depending on the context.
Line 54: (through the action of different factors) – “list few”
Line 55: Remove “+” after C3aR in this manuscript.
Line 62: Include “bacteria” along with viruses
Line 87: This statement “Local C biosynthesis and expression” contradicts the statement in line 47.
Line 95: Remove “; for review, see”
Line 102: Remove "26”
Line 107-108: Remove " (for comprehensive and seminal review,”
Line 125: Remove "pathway”
Line 126: Should be “in this study, we found that”
Line 129: “classical C3a”: what experiment was carried out to confirm this?
Line 137: Reference for “(see cohort study described in Brady et al. 2004)”
Line 138: Rewrite “more recently (2020-2022) at the teaching hospital (CHU) of La Reunion with reference”
Line 158: Dilution of antibodies used in this study?
Line 160: Is it “µg/ml or mg/ml”
Line 165-196: Remove paragraphs and combine the sentences to make immunostaining experiments in one paragraph.
Line 198: Should be “In vitro human cell lines” and remove “models”
Line 199: mention “subtypes”
Line 201-202: Specify which medium was medium was used for cell type
Line 203: should be “0.1”
Line 208: Remove “Recombinant human interferon-gamma (” and should be IFN-g, TGF-β1…. Were”
Line 209: Expand “BB”
Line 213: Remove “directly” and “culture”
Line 218: Is it equal quantity of RNA?
Line 222: Explain “probe”
Line 234: Remove “appearing”
Line 236: Remove “as we previously observed (data not shown)” and “in contrast”
Line 235: Should be “in all four”
Line 242: Specify “tumor” and remove “of note”
Line 246: Specify “(a), (b), (c), and (d)”
Line 246: Figure 1: using arrows point out “C3aR in 1a, “C3aR like cells” in 1b and similarly in figure 2, fig 3b and 3c, fig 4b and 4c, and fig 5.
Line 252-253: Figure 1 pictures represents which magnification?
Line 264: which “serial sections”?
Line 278: Figure 2: arrows representing C3aR, CD45, Cd68, CD31, SMA and CD1a in stained sections
Line 282: figure 2: Magnification used is missing
Line 310: Remove “atleast”
Line 329: also from “C4b2b of the Classical and Lectin pathway”
Line 330: Should be cathepsin B and L
Line 332: which type of “GBM”
Line 340: Figure 5: arrows representing blood vessel lumen and areas of tumor cell necrosis” and magnification of figures is missing.
Line 352-353: Rewrite “as yet through ill-characterized mechanisms”
Line 354: Remove “relevant”
Line 358: Remove “first”
Line 361: Remove “For example”
Line 364-367: IFN-g didn’t increase C3 and CCL5 expression in U251MG cells
Line 368: Figure 6: X-axis and y-axis labels are not clear
Line 371: What are “basal conditions (CT)” or is it untreated?
Line 372: Remove “(IFN)”
Line 374: figure 6: what is the “p-value “ for the results? Denote what is “#, ##, ### and ####” in legend.
Line 378: Should be “brain” instead of “CNS”
Line 381: which “GBM” type
Line 386-387: Specify the “several factors”
Line 404: Should be “factor B and factor D”
Line 410-412: “The CAF may also produce C3, as recently described in human melanoma and head and neck cancer using single- cell RNA sequencing experiments [34,57]” not related to the present study.
Line 413-417: The present study point out alternative complement pathway in generation of C3a where as in lines 413-417 highlight classical pathway role.
Line 417-419: rewrite “robust Bb staining ….for C3a”
Line 436-454: How does this relate to the present study
Line 481: Remove “)”
Line 493: Should be “tumor progression”
Reviewer 2 Report
Authors present an in - vitro study on specimens, i.e. paraffin blocks of 4 patients with GBM, 2 with diffuse astrocytoma, 2 oligodendrogliomas, 2 ependymomas WHO II and 2 metastases as well as normal brain tissue to analyzed the distribution and the regulated expression of C3a and its receptor C3aR in various primary and secondary brain tumor. The alternative complement pathway is activated to generate C3a. C3aR was upregulated in glioblastoma multiforme (GBM) and much less expressed in other brain tumors apart from diffuse astrocytoma; it was observed in tumor-associated macrophages (TAM) expressing CD68, CD18, CD163, and the proangiogenic VEGF. High levels of C3a were detected in the parenchyma of GBM. TGF-b1 was identified as one of the most potent growth factors that upregulate VEGF, C3, and C3aR in TAM (PMA-differentiated THP1) cell lines.
Major drawback of this study is low number of patients, i.e. specimens. Furthermore, this study ignores two important distinctive types of glioblastomas - IDH wildtype and mutated; and it is also possible to conduct this differentiation using novel kits also on specimens from the year 2004. Materials and Methods need to be reviewed by molecular biologist. Definition of "normal-appearing brain tissue" (Line 234) needs to be clarified - was the comparison conducted between "normal appearing cells" of a patient with a brain tumor with "glioma" cells of the brain tumor OR above mentioned cells of patients with brain tumors compared to specimens of patients without brain tumors? - OR is this comparison of current results to the study conducted to some published or non-published data which were here not presented? OR is this comparison a simple comparison to the data from the current literature? Before further review, this needs to be clarified.
If the comparison is with the current literature, this study is highly descriptive and the conclusions are questionable.
There are only 37 hits on complement activation and glioblastoma in PubMed, so this subject seems not to be so much in the midpoint of the research interest at the moment. I suggest to include a thorough literature review and discuss this issue.
Author Response
REVIEWER 2
Authors present an in - vitro study on specimens, i.e. paraffin blocks of 4 patients with GBM, 2 with diffuse astrocytoma, 2 oligodendrogliomas, 2 ependymomas WHO II and 2 metastases as well as normal brain tissue to analyzed the distribution and the regulated expression of C3a and its receptor C3aR in various primary and secondary brain tumor. The alternative complement pathway is activated to generate C3a. C3aR was upregulated in glioblastoma multiforme (GBM) and much less expressed in other brain tumors apart from diffuse astrocytoma; it was observed in tumor-associated macrophages (TAM) expressing CD68, CD18, CD163, and the proangiogenic VEGF. High levels of C3a were detected in the parenchyma of GBM. TGF-b1 was identified as one of the most potent growth factors that upregulate VEGF, C3, and C3aR in TAM (PMA-differentiated THP1) cell lines.
Major drawback of this study is low number of patients, i.e. specimens. Furthermore, this study ignores two important distinctive types of glioblastomas - IDH wildtype and mutated; and it is also possible to conduct this differentiation using novel kits also on specimens from the year 2004. Materials and Methods need to be reviewed by molecular biologist. Definition of "normal-appearing brain tissue" (Line 234) needs to be clarified - was the comparison conducted between "normal appearing cells" of a patient with a brain tumor with "glioma" cells of the brain tumor OR above mentioned cells of patients with brain tumors compared to specimens of patients without brain tumors? - OR is this comparison of current results to the study conducted to some published or non-published data which were here not presented? OR is this comparison a simple comparison to the data from the current literature? Before further review, this needs to be clarified.
If the comparison is with the current literature, this study is highly descriptive and the conclusions are questionable.
There are only 37 hits on complement activation and glioblastoma in PubMed, so this subject seems not to be so much in the midpoint of the research interest at the moment. I suggest to include a thorough literature review and discuss this issue.
Thank you for taking the time to review our manuscript. We appreciate your comments regarding the limitations of our study, and we have taken note of your suggestions for improvement.
We acknowledge that our study was limited by the small number of patient specimens included in the analysis. We are pleased to inform you that we have increased the number of patients in our study to 37, which includes 30 adult-type diffuse gliomas classified according to the latest WHO classification of CNS tumors. We believe that this larger sample size strengthens the significance of our findings.
Regarding the definition of "normal-appearing brain tissue," we apologize for any confusion that may have arisen from our description. In our study, we compared the expression levels of C3a and C3aR in brain tumor specimens to those in adjacent non-tumor brain tissue samples, which were histologically confirmed to be free of tumor cells. We have revised the manuscript to clarify this point and including new figures.
We appreciate your suggestion to include a more thorough literature review and discussion of the current state of research on complement activation and GBM. We have revised the manuscript to provide a more comprehensive overview of the existing literature about the complement activation in GBM.
With regards to the expression and role of complement in cancer seminal studies have been carried out in human (as reviewed in Afshar-Kharghan, V. The Role of the Complement System in Cancer. J. Clin. Invest. 2017). Focusing on C3aR several important and more recent contributions have been published in high impact journals and essentially making use of C3aR knockout mice: Davidson, S.; Efremova, M.; Riedel, A.; Mahata, B.; Pramanik, J.; Huuhtanen, J.; Kar, G.; Vento-Tormo, R.; Hagai, T.; Chen, X.; et al. Single-Cell RNA Sequencing Reveals a Dynamic Stromal Niche That Supports Tumor Growth. Cell Reports 2020, 31, 107628, doi:10.1016/j.celrep.2020.107628. Consonni, F.M.; Bleve, A.; Totaro, M.G.; Storto, M.; Kunderfranco, P.; Termanini, A.; Pasqualini, F.; Ali, C.; Pandolfo, C.; Sgambelluri, F.; et al. Heme Catabolism by Tumor-Associated Macrophages Controls Metastasis Formation. Nat. Immunol. 2021, 22, 595-+, doi:10.1038/s41590-021-00921-5. Magrini, E.; Di Marco, S.; Mapelli, S.N.; Perucchini, C.; Pasqualini, F.; Donato, A.; Guevara Lopez, M. de la L.; Carriero, R.; Ponzetta, A.; Colombo, P.; et al. Complement Activation Promoted by the Lectin Pathway Mediates C3aR-Dependent Sarcoma Progression and Immunosuppression. Nat. Cancer 2021, 2, doi:10.1038/s43018-021-00173-0.
In our study we are providing novel data regarding the expression of C3aR in human brain cancers and backing up the mouse data aforementioned , ie elevated C3aR expression together with C3a may contribute to the polarization of TAM and contributing to angiogenesis (VEGF).
We hope that our revisions have addressed all of your concerns, and we look forward to your favorable consideration of our manuscript for publication.

Reviewer 3 Report
The paper evaluates expression of the complement anaphylatoxin C3a and its receptor C3aR in the context of primary and metastatic brain tumors, with the aim of identification of new targets for these types of tumors. The study is performed on FFPE tissue slides and brain tumors/macrophages cell lines, identifying expression of C3a and C3aR in both tumors and macrophages, which could play an important role in VEGF secretion in tumor microenvironment.
The manuscript is clear and relevant for the field, being presented in a well-structured manner. The experimental design is appropriate to test the hypothesis. The manuscript results are reproducible based on the details provided in the methods section. The methods used are quite complex, from immunostaining experiments on FFPE tissues to cell cultures, IF and qRT-PCR on in vitro stimulated cells.
The figures containing graphs and IHC/IF images are appropriate and show the data properly, being easy to understand.
This is an important study on the role of C3a/C3aR in development of brain tumors, providing evidence for angiogenesis induced by the presence of these complement system components. The conclusions are consistent with the evidence and arguments presented. The cited references are relevant publications and do not include self-citations. Although I am not qualified to judge the English language, maybe the English proofreading will improve the overall quality of the paper.
Author Response
REVIEWER 3
The paper evaluates expression of the complement anaphylatoxin C3a and its receptor C3aR in the context of primary and metastatic brain tumors, with the aim of identification of new targets for these types of tumors. The study is performed on FFPE tissue slides and brain tumors/macrophages cell lines, identifying expression of C3a and C3aR in both tumors and macrophages, which could play an important role in VEGF secretion in tumor microenvironment.
The manuscript is clear and relevant for the field, being presented in a well-structured manner. The experimental design is appropriate to test the hypothesis. The manuscript results are reproducible based on the details provided in the methods section. The methods used are quite complex, from immunostaining experiments on FFPE tissues to cell cultures, IF and qRT-PCR on in vitro stimulated cells.
The figures containing graphs and IHC/IF images are appropriate and show the data properly, being easy to understand.
This is an important study on the role of C3a/C3aR in development of brain tumors, providing evidence for angiogenesis induced by the presence of these complement system components. The conclusions are consistent with the evidence and arguments presented. The cited references are relevant publications and do not include self-citations. Although I am not qualified to judge the English language, maybe the English proofreading will improve the overall quality of the paper.
Thank you for your positive feedbacks on our manuscript. We are glad to know that our study is considered important for the field, and we thank you for acknowledging the evidence we have provided on the role of C3a/C3aR in the development of brain tumors.

Round 2
Reviewer 2 Report
Authors have sufficiently responded to reviewer remarks.